# Food Is Medicine: Diet Assessment Tools in Adult Inflammatory Bowel Disease Research

**DOI:** 10.3390/nu17020245

**Published:** 2025-01-10

**Authors:** Vibeke Andersen, Anette Liljensøe, Laura Gregersen, Behrooz Darbani, Thorhallur Ingi Halldorsson, Berit Lilienthal Heitmann

**Affiliations:** 1Molecular Diagnostics and Clinical Research Unit, Department of Internal Medicine, University Hospital of Southern Denmark, 6200 Åbenrå, Denmark; a.liljensoe@rsyd.dk (A.L.); laura.gregersen@rsyd.dk (L.G.); bds@rsyd.dk (B.D.); tih@hi.is (T.I.H.); 2Institute of Regional Health Research, University of Southern Denmark, 5000 Odense, Denmark; 3Faculty of Food Science and Nutrition, University of Iceland, 101 Reykjavik, Iceland; 4Research Unit for Dietary Studies at The Parker Institute, Bispebjerg and Frederiksberg Hospital, 2000 Frederiksberg, Denmark; berit.lilienthal.heitmann@regionh.dk; 5Section for General Medicine, The Department of Public Health, University of Copenhagen, 1353 Copenhagen, Denmark

**Keywords:** inflammatory bowel disease, diet, diet assessment, food, diet quality, women, sex, gender, chronic disease, dietary indices

## Abstract

Background: Diet significantly impacts the onset and progression of inflammatory bowel disease (IBD), and diet offers unique opportunities for treatment and preventative purposes. However, despite growing interest, no diet has been conclusively associated with improved long-term clinical and endoscopic outcomes in IBD, and evidence-based dietary guidelines for IBD remain scarce. This narrative review critically examines dietary assessment methods tailored to the unique needs of IBD, highlighting opportunities for precision and inclusivity. Methods: We conducted a comprehensive literature review using search terms related to diet, diet assessment, nutrition, food, sex, gender, equity, and IBD. Results: The identified dietary assessment tools evaluated nutritional quality, dietary patterns, food processing, lifestyle interactions, inflammatory potential, and effects of specific nutrients. Advanced methods, including biomarkers, multi-omics approaches, and digital tools, were highlighted as being complementary to traditional approaches, offering enhanced precision and real-time monitoring. Women remain under-represented in dietary research but face unique nutritional needs due to hormonal cycles, pregnancy, and higher malnutrition risks in IBD. Discussion: Traditional diet assessment methods remain valuable but are often limited by misreporting biases. Advanced approaches may provide greater precision, enabling real-time monitoring and personalised dietary tracking. Incorporating considerations of sex, gender, age, ethnicity, socioeconomic, and sustainability enhances the relevance and applicability of these methods. Addressing these multifaceted aspects of dietary assessment in IBD can facilitate robust interventional trials. Conclusions: Diet assessment tools are essential for developing personalised dietary interventions in IBD, informing evidence-based guidelines, and improving health outcomes and quality of life in IBD.

## 1. Introduction

The global prevalence of prominent chronic conditions is rising alarmingly, leading to a significant healthcare burden, and diet is increasingly recognised as a key factor influencing the onset and progression of many chronic diseases [1]. According to the Global Burden of Disease Study, dietary risk factors contribute to 11 million deaths and 255 million disability-adjusted life years annually [2]. In addition, data from three US cohorts followed for up to 32 years demonstrated that adherence to a healthy diet is associated with a lower risk of major chronic diseases [3]. Finally, dietary intervention studies have demonstrated that diet can effectively treat or delay some diseases [4]. Such studies foster the concept of food as medicine.

Inflammatory bowel diseases (IBDs), encompassing Crohn’s disease (CD) and ulcerative colitis (UC), are chronic conditions of the gastrointestinal tract [5,6]. Recent advances in gut microbiome research have highlighted the interplay between diet, the gut microbiome, and the gut immune system as critical factors in modulating IBD processes [7]. This has led to a large number of recent publications on the potential role of diet in IBD onset and progression [8,9,10,11,12,13,14,15,16,17,18]. Moreover, patients and health professionals are increasingly aware of diet’s potential therapeutic role and demand dietary guidance. However, despite growing interest, no diet has been conclusively associated with improved long-term clinical and endoscopic outcomes in IBD, and evidence-based dietary guidelines for IBD remain scarce [19].

Diet interacts intricately with various biological, social, and environmental factors. Notably, sex and gender significantly influence dietary needs and responses, yet these aspects are often overlooked. Limited female representation in nutritional studies, coupled with a lack of sex-specific analyses, hampers our understanding of how diet affects men and women differently [20,21]. The inclusion of sex and gender in research enhances the validity, relevance, and applicability of findings [22]. Addressing these gaps is essential for creating nutritionally adequate and contextually relevant dietary guidance.

Moreover, the interplay between food and climate change underscores the importance of sustainable dietary practices [23,24]. Traditional diets, often richer in diverse and nutrient-dense foods, are generally healthier than the more modern Western diets, which tend to be highly processed and high in fat, sugar, and salt.

Understanding how diet influences disease mechanisms is crucial for harnessing diet as a therapeutic tool in IBD management [25]. To address this, we need to identify the key dietary components and identify the appropriate methods for dietary assessment in IBD. This review critically examines dietary assessment methods relevant to the unique needs of IBD, with a focus on addressing sex differences.

## 2. Methods

### Search Strategy and Selection Criteria

The research question for this narrative review is concerned with “dietary assessment methods relevant to the unique needs of IBD, with a focus on addressing sex differences”. Repeated searches on PubMed, EMBASE, and Cochrane identified data up till December 2024 for this review. The search terms were used alone or combined: food, nutrition, chronic disease, sex, gender, women, diet, diet assessment, diet quality, dietary indices, lifestyle, Crohn’s disease, inflammatory bowel disease, ulcerative colitis, and personalised/personalised. In addition, backward citation searches were conducted on the included references. Only papers published in English were reviewed. The final reference list was generated based on quality, originality, and relevance to the broad scope of this review.

## 3. Results

### 3.1. Diet Is Linked to IBD

The aim of this review was not to examine the effects of diet on the risk of IBD or its disease course, as these topics have been the focus of many recent excellent publications. These studies have highlighted that no diet has been conclusively associated with improved long-term clinical and endoscopic outcomes in IBD [8,9,10,11,12,13,14,15,16,17,18].

The link between diet and health in IBD has long been recognised. Patients with IBD are at significant risk of malnutrition due to various factors. Many patients report avoiding certain foods to prevent disease relapse, particularly during active disease phases, which further heightens the risk of malnutrition, as documented in multiple studies [14,26,27]. In addition to food avoidance, impaired nutrient absorption, intestinal loss, and increased nutritional demands caused by inflammation can contribute to malnutrition.

Interestingly, diet has also been demonstrated to play a therapeutic role in IBD. Exclusive enteral nutrition, also referred to as enteral feeding, is a type of treatment that includes a nutritional formula taken by mouth or by placing a tube in the nose or in the gut. It is a well-established treatment for active CD, especially in children [8,9,14].

Research on diet and dietary therapies has grown substantially over the past decade as prospective studies suggest that specific food items may be associated with the risk of developing IBD and other studies propose a role of diet in maintenance therapy [14,28,29]. This insight is reflected in the increasing number of dietary intervention studies in IBD, and many more are registered on ClinicalTrials.gov [8]. In particular, the role of ultraprocessed food has recently been investigated in IBD [13]. However, despite this growing interest, the evidence on diet’s effects on IBD remains limited [30,31]. A systematic review and meta-analysis of the most rigorous dietary trials concluded that the evidence supporting specific dietary interventions is generally of low or very low quality [30].

These studies further highlight the need for validated methods to assess different aspects of diet in IBD.

### 3.2. Diet and Under-Represented Groups

In addition to the biological complexities of IBD, dietary behaviours and nutritional needs are further shaped by sex, gender, and socioeconomic and cultural factors. A large Italian study found that many men prefer red and processed meat, with significantly higher consumption rates than women. Women, on the other hand, show a greater inclination towards consuming vegetables, whole grains, tofu, and high-cocoa-content dark chocolate, aligning with healthier food choices [32,33]. Further, biological factors like pregnancy and hormonal cycles create unique nutritional requirements. Consequently, iron requirements are higher in females due to menstrual losses [20].

Emerging studies suggest that sex differences may also influence physiological responses to various dietary patterns [20,21]. For instance, a study reported that men had higher postprandial triglyceridemia than women on a high-fat high-cholesterol diet, whereas no difference was observed on a low-fat low-cholesterol diet [20]. Further, a Western-style diet (high-fat/low-fibre) was found to alter the microbial profile differently in males and females, with females showing higher levels of Campylobacter and other species [21]. Additionally, another study found that a Mediterranean diet induces more favourable changes in glucose and insulin homeostasis in men compared to women [21]. However, evidence in humans is still sparse.

These multidimensional influences highlight the need for personalised and inclusive dietary research to inform effective nutritional strategies in IBD management.

### 3.3. Data Collection

#### 3.3.1. Asking Patients About Their Diet

Traditionally, dietary data have been obtained by asking participants about their past food intake.

Food Frequency Questionnaires (FFQs):

FFQs are self-administered tools, which capture how often a smaller or a larger number of specific foods are consumed over a defined period (e.g., daily, weekly, or monthly). They typically include a list of foods with varying portion sizes, allowing participants to select both the frequency and the portion size that best matches their intake of each food.

24-Hour Dietary Recalls:

Participants report all food and beverages consumed in the past 24 h, either through interviews or self-administered tools. For total diet assessment, 2–3 times of 24 h recalls are often used.

Diet History Interviews:

These are performed by a dietician or nutritionist and involve detailed discussions about usual food intake over an extended period (e.g., the past month or year), assisted by cups, measurements, and photos of portion sizes and preparation methods to better quantify intake.

#### 3.3.2. Real-Time Diet Recording

Alternatively, dietary intake can be recorded in real time.

Food Diaries:

Participants document all foods and beverages consumed over several days (typically 3 to 7), including portion sizes, types, and preparation methods.

With advancements in technology, digital tools such as mobile apps and wearable devices are increasingly employed to track dietary intake. These tools enable participants to log meals, capture images of food, or monitor specific dietary aspects. While convenient and widely accepted, digital methods share common limitations with traditional approaches, including recall and social desirability bias leading to misreporting.

### 3.4. Dietary Indices

Dietary assessment often involves quantifying and qualifying total diet and nutrient intake based on participant information, using various indices that offer insights into nutritional quality, environmental impact, and disease relevance. Examples and their application to IBD are summarised in Table 1 [23,34,35,36,37,38,39,40,41,42,43,44,45].

#### 3.4.1. Nutritional Quality Indices

FSAm-NPS Dietary Index (FSAm-NPS-DI):

This index, developed by the UK Food Standards Agency, assesses the nutritional quality of diets by classifying food products into five categories (A–E), with A indicating the highest nutritional quality. Individual FSAm-NPS-DI scores are calculated by averaging the scores of all foods consumed [34].

Planetary Health Diet:

Defined by the EAT-Lancet Commission, this evidence-based diet promotes health and sustainability by emphasising vegetables, fruits, whole grains, legumes, nuts, and unsaturated oils, with minimal red meat, processed meat, added sugars, and refined grains [23].

#### 3.4.2. Dietary Patterns

Dietary patterns focus on combinations of foods and beverages typically consumed by populations or cultures.

Mediterranean and Nordic Diets:

These emphasise whole, minimally processed foods, plant-based components, healthy fats, and moderate alcohol intake, often incorporating local and seasonal products.

Western Diets:

These are characterised by a high intake of processed foods, animal products, unhealthy fats, and added sugars, with less emphasis on cultural or geographical traditions.

Some of these tools have been used to assess dietary quality in IBD [37,45].

#### 3.4.3. Food Processing and Lifestyle Interaction

NOVA Classification System:

This system categorises food by processing levels into four groups, from minimally processed to ultraprocessed. Ultraprocessed foods often contain industrial ingredients (e.g., maltodextrin, hydrogenated oils, and cosmetic additives like dyes and emulsifiers) and are associated with negative health outcomes [36,60].

Diet and Lifestyle:

Risk scores have been developed to combine key dietary and lifestyle factors into comprehensive measures [38,39]. These scores are calculated based on the presence or levels of these factors and are used to categorise study participants into distinct groups. For instance, the Healthy Lifestyle Score incorporates data on the consumption of fruits, vegetables, fish, whole grains, refined grains, processed meats, and unprocessed red meats, along with information on smoking status, BMI, sleep quality, physical activity, and alcohol intake [39]. The modifiable risk score integrates dietary factors such as fruit, vegetable, red meat, and fibre intake, the ratio of n3 to n6 polyunsaturated fatty acids, with information on BMI, family history of IBD, history of appendectomy, physical activity, smoking status, and the use of non-steroidal anti-inflammatory drugs (NSAIDs) [38].

#### 3.4.4. Inflammatory Potential of Diet

Empirical Dietary Inflammatory Pattern (EDIP):

EDIP scores classify 18 food groups based on their association with inflammatory biomarkers and are often used in large cohort studies. Higher scores indicate a pro-inflammatory diet, while lower scores suggest an anti-inflammatory diet [35].

Dietary Inflammatory Index (DII):

DII scores are derived from an extensive literature review linking dietary components to inflammation markers. Unlike EDIP, DII includes a broader range of foods, nutrients, and bioactive compounds, assigning scores to items like processed meats, refined carbohydrates, and sugary foods [54].

#### 3.4.5. Specific Nutrients and Food Items

Individual components, such as dietary fibre and meat, have been studied for their potential roles in IBD [40,41,61,62]. For example, the plant-to-animal protein ratio was developed to investigate the role of the source of protein [57].

### 3.5. Molecular Markers

An alternative approach for exploring the connection between diet and health outcomes involves using molecular markers tailored to the investigated factor (Table 1). Biomarkers of food and nutrient intakes—derived from laboratory data, epigenetics, and metabolomics—provide objective measures of nutrients [63,64]. While these biomarkers have been linked to dietary intake, smoking behaviour, and key biological mechanisms, their predictive power remains modest. Despite their potential to enhance our understanding of diet–disease relationships, biomarkers are not yet widely applied in nutritional epidemiology [65].

### 3.6. Precision Nutrition

Precision nutrition is a rapidly growing field focusing on the integration of diet with genetic, metabolic, behavioural, and sociocultural factors to address inter-individual variability in health responses (Table 1) [66,67]. Initiatives such as the US Nutrition for Precision Health Initiative (NPH, https://nutritionforprecisionhealth.org/) and the Danish Precision Health Initiative (DELPHI) aim to advance this field by leveraging artificial intelligence (AI) and machine learning. These programs strive to improve our understanding of diet-related biological and physiological variability, offering the potential for more personalised nutritional strategies.

## 4. Discussion

Although the link between diet and health in IBD is well established, we still lack a clear understanding of how to optimise dietary interventions and effectively guide patients. A key step in addressing this challenge is improving our approach to diet assessment.

Ideal tools should focus on food items particularly relevant to patients with IBD, such as those that help maintain remission and address issues like malnutrition and food avoidance, especially during active disease phases. Additionally, they should account for factors like gender, age, and socioeconomic status, which influence food choices and nutrient intake.

The key aspects of diet assessment are outlined below.

### 4.1. Strengths, Limitations, and Gaps of Dietary Assessment Tools

Comparability across various conditions and studies may be a consideration. Widely used tools such as the FSAm-NPS-DI, the planetary health diet, NOVA classification, and Mediterranean diet scores have been applied in large-scale studies to assess the risk of various health conditions, including all-cause mortality [46,68,69,70,71,72,73,74]. Their validation ensures reliability and facilitates comparability across studies.

Many previous studies on diet and IBD have focused on specific nutrients or foods, such as protein or red meat intake, to understand the specific effects of individual nutrients. Nonetheless, this method has limitations, as it may overlook interactions between nutrients, foods, and other confounding factors. Investigating dietary patterns, rather than individual nutrients or foods, provides a more comprehensive view of the combined effects of various nutritional factors within a diet.

Further, incorporating lifestyle factors into dietary assessments can be particularly relevant for diseases like IBD, where these factors play a critical role. For instance, smoking status is especially impactful in IBD research, given its significant effect on disease risk and progression. However, combining dietary and lifestyle factors often involves assigning equal weight to each factor, which may not accurately reflect their varying levels of importance in influencing disease outcomes [38,39].

### 4.2. Data Collection

Traditionally, assessment of habitual diet intake relies on self-reporting using methods such as FFQs or 24 h dietary recalls in most larger studies. Clinical studies may also use dietary records or dietitian evaluations of diet histories. All these methods are prone to misreporting biases due to participants’ memory, generalisation ability, or willingness to provide accurate information. Prospective studies, where dietary data are collected before the outcome is known, reduce the consequences of misreporting biases because they are expected to be equal (random) across the groups. These biases lead to large data variability, which reduces the power to detect differences between groups. However, this is not always the case as people with obesity, women, adolescents, or those from high socioeconomic positions are more prone to social desirability bias, and children, older individuals, or people with dementia are hampered by memory issues and may have difficulties in generalising what they eat [75]. Also, it has been demonstrated, using dietary recovery biomarkers, that most foods forgotten or omitted in diet reporting are rich in fat and/or carbohydrate, while protein reporting is quite accurate [76]. The consequence of such non-random biases can be both inflated and attenuated results [77].

Dietary information is often collected only once in studies, limiting insights into long-term dietary changes. However, emerging technologies offer opportunities for improvement. Wearable devices enable real-time dietary monitoring, although they can still be cumbersome for users. Mobile apps and digital tools, including image-based food recognition systems, provide scalable and cost-efficient solutions for dietary assessment [78,79]. These technologies allow for long-term continuous tracking, instant feedback, and integration with clinical data, enabling tailored interventions to gain comprehensive insights into the relationship between diet and individual health outcomes and empower patients to understand the relationship between their diet and health outcomes.

### 4.3. Diet and Under-Represented Groups

Despite considerable under-reporting, the emerging literature suggests that nutritional requirements, dietary preferences, and responses to diet differ between men and women [20,21,32]. However, women are under-represented in IBD research, particularly elderly women, women of colour, and the socioeconomically underprivileged, a phenomenon called intersectionality [22]. This limits biological understanding and contributes to health inequities and social injustice. The under-representation of certain groups (e.g., women, the elderly, and individuals from diverse ethnic and socioeconomic backgrounds) reduces the generalisability of findings. For example, it is likely that important outcomes have previously been missed as sex differences had not been accounted for. Hence, addressing sex and gender, age, ethnicity, geography, and socioeconomic status in research will improve the validity and applicability of research within IBD.

### 4.4. Diet Interactions and Food Content Variability

The effects of diet on IBD may be further shaped by individual factors such as genetics and gut microbiome composition [80,81]. For example, patients with impaired gut microbial fermentative capacity metabolise certain diets, such as those rich in carbohydrates, differently than healthy individuals [7,82]. In IBD patients, who typically have low gut microbiome diversity, these variations are particularly relevant.

Diet may also be influenced by drugs. For example, long-term use of proton pump inhibitors may lead to decreased absorption of dietary calcium [83]. In addition, nutrients also interact with one another, affecting their digestion and absorption. For example, vitamin C aids in non-heme iron uptake, vitamin D enhances calcium absorption, and soluble fibre slows carbohydrate absorption [84,85].

Variation in food content adds another layer of complexity. As reported by the Food and Agriculture Organisation of the United Nations, food composition changes with seasonal and geographical variations, and preparation methods (e.g., cooking) can alter nutrient levels, such as resistant starch, which may be particularly relevant for IBD patients (https://www.fao.org/4/y4705e/y4705e.pdf, accessed on 27 December 2024).

These interactions, along with residual confounding, complicate efforts to establish causality between diet and health outcomes, including IBD. A systems biology approach, enabled by multi-omics profiling, can help account for these factors and deepen our understanding of diet biology [66].

### 4.5. Biomarkers and Multi-Omics Analysis

Food intake biomarkers and multi-omics precision nutrient analyses offer objective methods to assess food intake, bypassing many inaccuracies linked to self-reported data. These approaches provide a more precise and reliable way to validate dietary data, investigate diet–health relationships, and understand disease mechanisms. Such progress has led to the reporting of strong correlations between diet metabolites and some food items such as coffee, citrus, alcohol, dairy, and broccoli after 2 weeks of a controlled diet [65]. However, there are challenges with biomarkers, including accuracy, reproducibility, and complex interpretation [67]. For instance, variation between individuals in biomarker absorption and excretion, interactions with gut microbes, intake frequency, and sampling timing can influence biomarker analyses [64]. Multi-omics studies can help characterise such factors. Accordingly, a recent study highlighted the complex interaction between host factors such as gut transit time and pH, fibre and protein intake, gut microbiome composition and metabolism, as well as the intra- and inter-individual differences, in a healthy cohort [86]. Consequently, advanced studies can assess and validate food intake biomarkers—for example, their usability for compliance assessment (classifying consumers versus non-consumers) or evaluating diets in epidemiological research (quantifying the intake of specific food or diet patterns)—before they are implemented in clinical practice [64]. Despite these challenges, biomarkers of food intake and multi-omics represent a promising advancement in dietary research.

### 4.6. Causality

Establishing causal relationships between diet and health is inherently challenging due to nutrient interactions, residual confounding, and the complexity of dietary intake. For example, ultraprocessed foods are associated with markers of poor diet quality (e.g., high added sugar, sodium, and low fibre), but it remains unclear whether these foods directly increase health risks or simply indicate overall poor diet quality [36,73,87].

Clinical trials are essential for demonstrating causality but are resource-intensive and, therefore, less frequently used. This leads to significant gaps in knowledge. Advanced methods, such as the integration of biomarkers, multi-omics analyses, and real-time digital dietary assessments, offer promising avenues for addressing these challenges and improving the reliability of causal research.

## 5. Conclusions and Future Directions

This review highlights the critical role of diet assessment tools in addressing the complexities of dietary intake in IBD. Accurate and reliable tools are essential for capturing factors unique to IBD, such as malnutrition risks, diet safety, food avoidance behaviours, and individual biological variability.

Here, we identified and reviewed methods for evaluating nutritional quality, dietary patterns, food processing, lifestyle interactions, inflammatory potential, and the effects of specific nutrients. Advanced assessment methods, including biomarkers, multi-omics technologies, and digital tools, provide opportunities to complement traditional approaches and improve precision. Emerging technologies, such as wearable devices and mobile apps, offer the potential to enhance real-time data collection and support personalised dietary monitoring. Women face unique nutritional challenges due to hormonal cycles, pregnancy, and higher malnutrition risks in IBD and remain under-represented in dietary research. In the end, the final decision on the most applicable dietary assessment tool for clinical practice or future research will depend on the local resources and the purpose. However, we recommend involving patient partners in the selection of the diet assessment tool to ensure its relevance for patients and to include objective diet measures.

Dietary assessment tools can support robust interventional trials and improve the precision of dietary assessments. Integrating considerations such as sex and gender, age, ethnicity, geography, and socioeconomic factors into studies of diet–IBD relationships enhances the relevance and applicability of research findings. Further, incorporating principles of equality and global sustainability enhances the usability of diet assessments for IBD by ensuring that dietary recommendations and interventions are relevant, equitable, and ethically responsible. This multidimensional approach acknowledges the broader implications of dietary choices on social, economic, and environmental systems, making them more applicable and impactful across diverse populations. In addition, the successful implementation of such outcomes further requires addressing factors such as the affordability of healthy foods, accessibility, user-friendliness, and adaptability to diverse populations.

In summary, diet assessment tools are indispensable for developing personalised dietary strategies tailored to the unique needs of patients with IBD. They inform evidence-based interventions and hold the potential to improve health outcomes, ultimately enhancing the quality of life for individuals living with IBD.

## Figures and Tables

**Table 1 nutrients-17-00245-t001:** Diet assessment tools for adult IBD research along with some examples of use.

Diet Assessment Tool		Use in IBD	
Nutrition Quality Indices			
FSAm-NPS DI score, Modified UK Food Standards	2018, Deschasaux M, PLoS Med [46]	Nutritional quality was associated with risk of CD but not UC in a prospective study of 394,255 participants.	2024, Meyer A, APT [34]
Planetary health diet, EAT-Lancet Commission	2019, Willett, Lancet [23]	A diet of plant-based proteins, unsaturated fats, whole grains, and ample fruit and vegetables promoted well-being and lowered the risk of developing major chronic diseases, as did limiting meat, refined grains, and sugar intake.	
Healthy Eating Index	2018, Krebs-Smith SM, J Acad Nutr Diet [47]	No association with risk of CD and UC among the 125,445 participants of the LifeLines Cohort Study.	2015, Krebs-Smith SM, Acad Nutr Diet [48]
Dietary Patterns			
Mediterranean vs. Western diet pattern	2011, Bach-Faig A, Public Health Nutr [49]	A randomised controlled trial reporting improved simple clinical colitis activity index scores and faecal calprotectin among the Mediterranean diet pattern group (15 participants) compared to the Canadian habitual diet pattern group (13 participants).	2023, Haskey, JCC [37]
Mediterranean diet	2003, Trichopoulou A, NEJM [50] 1995, Trichopoulou A, BMJ [51]	Greater adherence to a Mediterranean diet was associated with lower risk of later-onset CD among 83,147 participants from the Cohort of Swedish Men and Swedish Mammography Cohort.	2020, Khalili H, Gut [43]
Nordic diet	2023, Blomhoff, NNR [45]		
Food Processing and Lifestyle Interaction			
NOVA Food Classification System	2018, Monteiro CA, Public Health Nutr [52]	Ultraprocessed food intake was associated with the risk of CD but not UC.	2024, Lane MM, BMJ [36] 2022, Lo CH, CGH [53] 2023, Narula N, CGH [42]
Weighted healthy lifestyle scores	2023, Sun Y, Am J Gastroenterol [39]	Having a favourable lifestyle reduced the risk of CD and UC considerably compared with those with an unfavourable lifestyle.	2023, Sun Y, Am J Gastroenterol [39]
Modifiable risk scores	2022, Lopes EW, Gut [38]	Population-attributable risks were calculated in US and European cohorts. Adherence to low modifiable risk scores (BMI, smoking, NSAID use, physical activity, intake of fruit and veg, fibre, n3:n6 PUFA, and red meat) could have prevented 4 in 10 CD and UC cases.	2022, Lopes EW, Gut [38]
Inflammatory Potential of Diet			
Empirical dietary inflammatory pattern (EDIP)	2016, Tabung FK, J Nutr [54]	EDIP was associated with a higher risk of CD but not UC among 166,903 women and 41,931 men in the Nurses’ Health Study and Health Professionals Follow-up Study.	2020, Lo CH, Gastroenterology [44]
Dietary inflammatory index (DII)	2014, Shivappa N, Public Health Nutr [55]	EDIP and DII were not associated with IBD incidence and progression among 121,472 participants from the UK Biobank.	2024, Wellens J, IBD [35]
Specific Nutrients and Food Items			
Meat		Meta-analysis suggested that each 100 g/d increment in dietary total meat consumption was associated with a 38% greater risk of IBD.	2023, Talebi S, Adv Nutr [41]
Dietary fibre intake	2018, Bradbury KE, J Nutr Sci [56]	Higher consumption of dietary fibre was associated with a lower risk of IBD and CD, but not UC among 470,669 participants of the UK Biobank study.	2023, Deng M, APT [40]
Plant-to-animal protein ratio	2017, Møller G, Nutrients [57]	No association with risk of CD and UC among 125,445 participants of the LifeLines Cohort Study.	2022, Peters V, JCC [48]
FODMAP	2022, Gibson PR, Eur J Nutr [58]	A low-FODMAP diet may improve clinical outcomes in the management of IBD and quality of life for patients, but concerns remain as to the adequacy of the diet.	2022, Simões CD, E J Nutr [59]

Abbreviations: BMI, body mass index; CD, Crohn’s disease; DII, dietary inflammatory index; EDIP, empirical dietary inflammatory pattern; NSAID, non-steroid anti-inflammatory drug; UC, ulcerative colitis.

## Data Availability

No datasets were generated or analysed as part of this study.

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
