# Peer review of "Food Is Medicine: Diet Assessment Tools in Adult Inflammatory Bowel Disease Research"

_nutrients, 2025, doi:10.3390/nu17020245_

Round 1
Reviewer 1 Report
Comments and Suggestions for Authors
This narrative review explores the role of diet and dietary assessment tools in the management and understanding of inflammatory bowel disease (IBD). It highlights traditional and advanced methods for dietary evaluation, including biomarkers and multi-omics, while emphasizing the need for inclusivity and precision in dietary research. The review underscores gaps in evidence-based dietary guidelines and the underrepresentation of women in IBD dietary studies.
This is a very well-written and thorough review. It effectively conveys the significance of diet in IBD and the limitations of current dietary research methods. However, there are areas that could be expanded upon to strengthen the discussion further:
-
Long-Term Outcomes and Diet: The authors should explicitly state that no diet has been conclusively associated with improved long-term clinical and endoscopic outcomes in IBD. This is a critical point that could shape clinical applications and future research directions. Discussing this more comprehensively would enhance the review's impact.
-
Impact of Migration on Dietary Patterns and IBD Risk: The review could benefit from a discussion on how changes in dietary patterns following migration may influence the onset or exacerbation of IBD. This aspect is particularly relevant when considering the interplay between environment, diet, and disease: https://pubmed.ncbi.nlm.nih.gov/28524546/ https://pubmed.ncbi.nlm.nih.gov/33206912/ .
-
Safety and Precision of Dietary Interventions: While the review highlights precision approaches, it could delve deeper into their safety profiles and the potential challenges in implementing these advanced methods in clinical practice.
Author Response
Reviewer 1
Comments and Suggestions for Authors
C1
This narrative review explores the role of diet and dietary assessment tools in the management and understanding of inflammatory bowel disease (IBD). It highlights traditional and advanced methods for dietary evaluation, including biomarkers and multi-omics, while emphasizing the need for inclusivity and precision in dietary research. The review underscores gaps in evidence-based dietary guidelines and the underrepresentation of women in IBD dietary studies.
This is a very well-written and thorough review. It effectively conveys the significance of diet in IBD and the limitations of current dietary research methods. However, there are areas that could be expanded upon to strengthen the discussion further:
- Long-Term Outcomes and Diet: The authors should explicitly state that no diet has been conclusively associated with improved long-term clinical and endoscopic outcomes in IBD. This is a critical point that could shape clinical applications and future research directions. Discussing this more comprehensively would enhance the review's impact.
R1: Thank you very much for taking the time to review this manuscript. Please find the detailed responses below and the corresponding revisions/corrections highlighted/in track changes in the re-submitted files.
Thank you for pointing this out. We agree with this comment. Therefore, we have changed the sentence in the abstract and introduction from “However, despite growing interest, evidence-based dietary guidelines for IBD remain scarce, apart from the established role of exclusive enteral nutrition in Crohn's disease.” to “However, despite growing interest, no diet has been conclusively associated with improved long-term clinical and endoscopic outcomes in IBD and evidence-based dietary guidelines for IBD remain scarce.”.
- Impact of Migration on Dietary Patterns and IBD Risk: The review could benefit from a discussion on how changes in dietary patterns following migration may influence the onset or exacerbation of IBD. This aspect is particularly relevant when considering the interplay between environment, diet, and disease: https://pubmed.ncbi.nlm.nih.gov/28524546/ https://pubmed.ncbi.nlm.nih.gov/33206912/ .
R2: Thank you for pointing this out. Migration may change many factors in addition to diet. In respect of your comment, we have changed the introductory phrase in the section diet is linked to IBD from “The aim of this review was not to examine the effects of diet on the risk of IBD or its disease course, as these topics have been the focus of many recent excellent publications.” to “The aim of this review was not to examine the effects of diet on the risk of IBD or its disease course, as these topics have been the focus of many recent excellent publications. These studies have highlighted that no diet has been conclusively associated with improved long-term clinical and endoscopic outcomes in IBD.” We considered including the mentioned reference about migration but agreed not to because migration involves changing factors other than diet.
- Safety and Precision of Dietary Interventions: While the review highlights precision approaches, it could delve deeper into their safety profiles and the potential challenges in implementing these advanced methods in clinical practice.
R3:
Thank you for pointing this out. We agree. Therefore, we have changed: “Food intake biomarkers and multi-omics precision nutrient analyses offer objective methods to assess food intake, bypassing many inaccuracies linked to self-reported data. These approaches provide a more precise and reliable way to validate dietary data, investigate diet-health relationships, and understand disease mechanisms. However, biomarkers have challenges, including accuracy, reproducibility, and complex interpretation. For instance, individual variation in absorption and excretion, interactions with gut microbes, and sampling timing can complicate analyses. Despite these challenges, biomarkers represent a significant advancement in dietary research.” to
“Food intake biomarkers and multi-omics precision nutrient analyses offer objective methods to assess food intake, bypassing many inaccuracies linked to self-reported data. These approaches provide a more precise and reliable way to validate dietary data, investigate diet-health relationships, and understand disease mechanisms. Such progress has led to the reporting of strong correlations between diet metabolites and some food items such as coffee, citrus, alcohol, dairy and broccoli after 2 weeks of a controlled diet. However, biomarkers have challenges, including accuracy, reproducibility, and complex interpretation. For instance, variation between individuals in biomarker absorption and excretion, interactions with gut microbes, intake frequency, and sampling timing can influence biomarker analyses. Multi-omics studies can help characterise such factors. Accordingly, a recent study highlighted the complex interaction between host factors such as gut transit time and pH, fibre and protein intake, gut microbiome composition and metabolism, as well as the intra- and inter-individual differences, in a healthy cohort. Consequently, advanced studies can assess and validate food intake biomarkers – for example, their usability for compliance assessment (classifying consumers versus non-consumers) or evaluating diets in epidemiological research (quantifying the intake of specific food or diet patterns) - before they are implemented in clinical practice. Despite the challenges, biomarkers of food intake and multi-omics represent a promising advancement in dietary research.”
Reviewer 2 Report
Comments and Suggestions for Authors
This paper explores important and timely questions related to dietary assessment in the context of Inflammatory Bowel Diseases (IBD). The topic is highly relevant given the increasing focus on precision nutrition and the role of diet in managing chronic conditions. However, there are several areas where the manuscript could be improved to enhance clarity, organization, and overall impact.
Major
- Clarity and Organization of the Table:
The table in the manuscript is difficult to interpret due to its disorganized presentation. It would benefit from:
Clearer and more descriptive column headings.
Grouping related data into logical categories or sections.
Consistent formatting throughout (e.g., uniform font size, alignment).
Emphasizing key findings or comparisons directly relevant to IBD research.
- Emphasis and Focus:
The manuscript provides an extensive overview but lacks a clear prioritization of the most critical findings or insights. The authors should:
Highlight the dietary assessment tools most applicable to clinical practice or future research. Include discussions of specific strengths, limitations, and gaps of these tools in addressing IBD-related dietary challenges.
- Missing Considerations:
The manuscript acknowledges sex and gender differences but could further expand on:
The practical implications of these differences for dietary recommendations and tool development.How underrepresentation of certain groups (e.g., women, elderly, individuals from diverse ethnic and socioeconomic backgrounds) impacts the generalizability of findings.
Minor
- References Formatting:References should be placed before commas to adhere to MDPI format.
Verify the accuracy of all reference citations and ensure consistency in formatting.
Author Response
Reviewer 2
This paper explores important and timely questions related to dietary assessment in the context of Inflammatory Bowel Diseases (IBD). The topic is highly relevant given the increasing focus on precision nutrition and the role of diet in managing chronic conditions. However, there are several areas where the manuscript could be improved to enhance clarity, organization, and overall impact.
Major
- Clarity and Organization of the Table:
The table in the manuscript is difficult to interpret due to its disorganized presentation. It would benefit from:
Clearer and more descriptive column headings.
Grouping related data into logical categories or sections.
Consistent formatting throughout (e.g., uniform font size, alignment).
Emphasizing key findings or comparisons directly relevant to IBD research.
R1: Thank you very much for taking the time to review this manuscript. Please find the detailed responses below and the corresponding revisions/corrections highlighted/in track changes in the re-submitted files.
Thank you for pointing this out. We agree with this comment. Therefore, we have used Clearer and more descriptive column headings as well as grouped related data into logical categories or sections. Thus, we have changed Column headings to: “Diet Assessment Tool”, and “Use in IBD”. Moreover, now, a consistent format has been used.
- Emphasis and Focus:
The manuscript provides an extensive overview but lacks a clear prioritization of the most critical findings or insights. The authors should:
Highlight the dietary assessment tools most applicable to clinical practice or future research. Include discussions of specific strengths, limitations, and gaps of these tools in addressing IBD-related dietary challenges.
R2: Thank you for pointing this out. We agree with this comment. Therefore, we have changed the heading of the first section in the discussion from “Choice of Dietary Assessment Tool” to “Strengths, Limitations, and Gaps of Dietary Assessment tools“. Further, in the conclusion and Future Directions section, we have added, “In the end, the final decision on the most applicable dietary assessment tool for clinical practice or future research will depend on the local resources and the purpose. However, we recommend involving patient partners in the selection of the diet assessment tool to ensure the relevance for patients, and to include objective diet measures.”
- Missing Considerations:
The manuscript acknowledges sex and gender differences but could further expand on:
The practical implications of these differences for dietary recommendations and tool development.How underrepresentation of certain groups (e.g., women, elderly, individuals from diverse ethnic and socioeconomic backgrounds) impacts the generalizability of findings.
R3: Thank you for pointing this out. We agree with this comment. Therefore, we have changed the heading of section 4.3. Diet and Sex And gender to 4.3. Diet and underrepresented groups. We have also changed the sentence “Therefore, it is likely that important outcomes have previously been missed as sex differences have not been accounted for.” to “The underrepresentation of certain groups (e.g., women, elderly, individuals from diverse ethnic and socioeconomic backgrounds) reduce the generalisability of findings. For example, it is likely that important outcomes have previously been missed as sex differences have not been accounted for. ”
Minor
- References Formatting:References should be placed before commas to adhere to MDPI format.
Verify the accuracy of all reference citations and ensure consistency in formatting.
R4: Thank you for pointing this out. This has been done.
Reviewer 3 Report
Comments and Suggestions for Authors
Dear Redactors,
Thank you very much for the opportunity to revise the article “Food Is Medicine: Diet Assessment Tools in Adult Inflammatory Bowel Disease Research”.
The article is very interesting and well written.
I have just a few remarks.
I suggest Authors to indicate the most suitable assessment tool in this group of patients. I get the feeling that it wasn’t state clearly.
I also suggest describing in more details each questionnaires, how the data are collected, by who etc.
Thanks
Author Response
Reviewer 3
Dear Redactors,
Thank you very much for the opportunity to revise the article “Food Is Medicine: Diet Assessment Tools in Adult Inflammatory Bowel Disease Research”.
The article is very interesting and well written.
I have just a few remarks.
I suggest Authors to indicate the most suitable assessment tool in this group of patients. I get the feeling that it wasn’t state clearly.
I also suggest describing in more details each questionnaires, how the data are collected, by who etc.
Thanks
R1: Thank you very much for taking the time to review this manuscript. Please find the detailed responses below and the corresponding revisions/corrections highlighted/in track changes in the re-submitted files.
Thank you for pointing this out. We agree with this comment. Therefore, in the conclusion and Future Directions section, we have added, “In the end, the final decision on the most applicable dietary assessment tool for clinical practice or future research will depend on the local resources and the purpose. However, we recommend involving patient partners in the selection of the diet assessment tool to ensure the relevance for patients, and to include objective diet measures.”
In addition, we have been through the questionnaire section and clarified the collection of data, by whom, etc. We have added “FFQs are self-administered tools which capture how often a smaller or larger number of specific foods are …”.
Editor
(I) Ensure all references are relevant to the content of the manuscript.
(II) Highlight any revisions to the manuscript, so editors and reviewers can see any changes made.
(III) Provide a cover letter to respond to the reviewers’ comments and explain, point by point, the details of the manuscript revisions.
(IV) If the reviewer(s) recommended references, critically analyze them to ensure that their inclusion would enhance your manuscript. If you believe these references are unnecessary, you should not include them.
(V) If you found it impossible to address certain comments in the review reports, include an explanation in your appeal.
R1: Thank you for pointing this out. It has been done. Revisions have been highlighted by yellow, apart from all references, which have been placed before commas without being marked.
Round 2
Reviewer 2 Report
Comments and Suggestions for Authors
The authors successfully responded to the reviewers' questions.